# Gallic Acid-Based Hydrogels for Phloretin Intestinal Release: A Promising Strategy to Reduce Oxidative Stress in Chronic Diabetes

**DOI:** 10.3390/molecules29050929

**Published:** 2024-02-20

**Authors:** Roberta Cassano, Federica Curcio, Roberta Sole, Silvia Mellace, Sonia Trombino

**Affiliations:** Department of Pharmacy, Health and Nutritional Sciences, University of Calabria, 87036 Rende, Italy; roberta.cassano@unical.it (R.C.); federica.curcio@unical.it (F.C.); mellace.silvia@gmail.com (S.M.)

**Keywords:** hydrogel, phloretin, gallic acid, antioxidant activity, intestinal release

## Abstract

Diabetes mellitus (DM) is a metabolic disease characterized by hyperglycemia caused by abnormalities in insulin secretion and/or action. In patients with diabetes, complications such as blindness, delayed wound healing, erectile dysfunction, renal failure, heart disease, etc., are generally related to an increase in ROS levels which, when activated, trigger hyperglycemia-induced lesions, inflammation and insulin resistance. In fact, extensive cell damage and death occurs mainly due to the effect that ROS exerts at the level of cellular constituents, causing the deterioration of DNA and peroxidation of proteins and lipids. Furthermore, elevated levels of reactive oxygen species (ROS) and an imbalance of redox levels in diabetic patients produce insulin resistance. These destructive effects can be controlled by the defense network of antioxidants of natural origin such as phloretin and gallic acid. For this reason, the objective of this work was to create a nanocarrier (hydrogel) based on gallic acid containing phloretin to increase the antioxidant effect of the two substances which function as fundamental for reducing the mechanisms linked to oxidative stress in patients suffering from chronic diabetes. Furthermore, since the bioavailability problems of phloretin at the intestinal level are known, this carrier could facilitate its release and absorption. The obtained hydrogel was characterized using Fourier transform infrared spectroscopy (FT-IR). Its degree of swelling (a%) and phloretin release were tested under pH conditions simulating the gastric and intestinal environment (1.2, 6.8 and 7.4). The antioxidant activity, inhibiting lipid peroxidation in rat liver microsomal membranes induced in vitro by a free radical source, was evaluated for four hours. All results showed that gallate hydrogel could be applied for releasing intestinal phloretin and reducing the ROS levels.

## 1. Introduction

Diabetes mellitus is a metabolic disease characterized by a congenital (type I insulin-dependent) or an acquired (type II non-insulin-dependent) inability to transport glucose from the blood to the cells [1,2]. Chronic hyperglycemia, due to a deficiency in insulin secretion or insulin resistance, implies an increased production of reactive oxygen species ROS responsible for oxidative damage in cellular constituents, including DNA deterioration and peroxidation of proteins and lipids, and the activation of injury and cell death [3]. Specifically, OS pathways would represent a connection between the disruption of the glucoregulatory system and tissue damage [4]. Hyperglycemic conditions can lead to a severe imbalance between ROS generation and antioxidant defense in organs. The source of hyperglycemia-related ROS contains lipid or glucose autoxidation and reduced activity of antioxidant enzymes, including SOD, GPX and CAT, all of which are involved in the antioxidant defense of tissues under diabetic conditions [5]. Once insulin resistance develops, stress response pathways are activated, which increases the expression of regulatory cytokines. In fact, increased ROS can lead to the dephosphorylation of insulin receptors by inhibiting the PI3K-Akt signaling pathway, which in turn leads to a reduction in glucose transporter 4 (GLUT4) translocation for glucose absorption [6]. In this context, literature data suggest that gallic acid, a naturally occurring phenolic acid, possesses pharmacological antioxidant, anti-inflammatory and protective activities against serious metabolic disorders, including cardiovascular diseases and diabetes [7,8]. The phenolic hydroxyl groups of gallic acid are able to interact with other molecules capable of generating free radicals and mediating radical oxidation [9]. They may also inhibit certain oxidative enzymes involved in radical production such as advanced oxidation protein products (AOPP), malondialdehyde (MDA) and oxidized low-density lipoprotein (oxLDL) [10]. The combination with phloretin, another flavonoid with antioxidant properties, could enhance antioxidant action in a hyperglycemic environment where ROS concentration is high and activate the PI3K/AKT signaling cascade to increase GLUT4 translocation and expression, resulting in improved glucose consumption and tolerance [11,12,13].

Unfortunately, however, these compounds, particularly phloretin, show low bioavailability via the oral route, as they are poorly adsorbed in the intestine as they are degraded by microorganisms and/or intestinal enzymes, such as intestinal glucosidases, and easily excreted via the urinary tract [14,15]. To improve stability, absorption profile and to prolong drug effect, dosage forms such as nano and microemulsions, liposomes, etc., are used as innovative carriers in drug delivery [16,17]. In order to facilitate the intestinal release of phloretin, we considered developing a gallic acid-based hydrogel, which can swell rapidly by absorbing large amounts of water and undergoing changes in response to stimuli from the external environment [18,19]. In order to facilitate the intestinal release of phloretin, we thought of developing a hydrogel capable of swelling rapidly by absorbing large amounts of water and undergoing changes in response to stimuli from the external environment [20,21]. Specifically, gallic acid was derivatized by esterification into gallic acid diacrylate and used as a polymer for the realization of a hydrogel in which phloretin was encapsulated. The resulting material was characterized using Fourier transform infrared spectroscopy (FT-IR). Its degree of swelling at equilibrium (α%) was evaluated in fluids simulating the gastric and intestinal environment. The release of phloretin by the hydrogel was carried out under the same conditions exploited for swelling studies. The antioxidant activity consisting in the inhibition of lipid peroxidation in rat liver microsomal membranes induced in vitro by a source of free radicals, such as tert-butyl hydroperoxide (*t*-BOOH), was evaluated after the exposure of the gallate hydrogel, containing phloretin (Figure 1), to gastrointestinal environmental conditions. The obtained results showed that this hydrogel could be potentially useful to increase the intestinal phloretin release and to exert antioxidant activity in diabetic patients to reduce insulin resistance.

## 2. Results and Discussion

### 2.1. Preparation of the Hydrogel Based on Gallic Acid Diacrylate

In this work, hydrogel for the transport and release of phloretin was constructed. Chemical groups susceptible to radical polymerization were introduced into the gallic acid structure via Steglich esterification with acrylic acid in THF. The formation of diacrylate gallic acid was confirmed using FT-IR, ^1^H-NMR and GC-MS analyses. Gallic acid-based hydrogel was prepared starting from diacrylate gallic acid using a reverse phase suspension copolymerization with N, N-dimethylacrylamide comonomer. The reaction was started by using ammonium persulfate as the initiator system. Comparing the FT-IR spectrum of diacrylate gallic acid with that of the hydrogel (Figure 2), it was revealed that the double bond absorption peaks of the methacrylic group disappeared after hydrogel formation because of crosslinking. FT-IR (KBr) ν (cm^−1^): 3324 (-OH), 3261 (-OH), 3034 (-CH), 1780 (-C = O), 1739 (-C = O), 1710 (-C = O), 1626 (-C = C), 1261 (-CO), 985 (-CH), 905 (-CH). M/Z: 205 (100%), 277 (4%).^1^H-NMR (CD_3_OD) δ (ppm): 5.916 (2H, dd), 6.378 (2H, dd), 6.675 (2H, dd), 7.651 (2H, d), 12.74 (1H, bs). Yield: 70% (Figure 1).

### 2.2. Morphological Analysis

The morphological characterization of the hydrogel was performed using electronic scanning microscopy (SEM). In particular, as shown in Figure 3 the hydrogel exhibited a porous structure attributable to the presence of hydrophilic groups in the hydrogel, which facilitates the permeation of water or biological fluids and explains the high degree of swelling at pH 7.4.

### 2.3. Swelling Studies

The swelling behavior of hydrogel (α%) was evaluated at three different pHs (1.2, 6.8 and 7.4) and at predetermined time intervals (1 h, 2 h, 3 h, 4 h) (Table 1). The results indicated that the swelling degree at pH 7.4 was greater than that found at pH 1.2. This could be explained by the electrostatic repulsion between the polymer chains due to the increase in dissociated groups.

### 2.4. Release Studies from Hydrogel

The release tests were conducted in simulated gastric fluid (SGF, pH 1.2), simulated intestinal fluid (SIF, pH 6.8) and simulated colon fluid (SCF, pH 7.4), at different time intervals (1 h, 2 h, 3 h, 6 h, 12 h). From these studies, the ability of the hydrogel to release more phloretin into the intestinal environment at pH 7.4 emerged, thus allowing us to hypothesize the use of this material as a vehicle for phloretin in the intestine (Figure 4).

### 2.5. Determination of Total Phenolic Content

The content of phenols was traced through the Folin–Ciocalteu test, by analysis using the UV–Vis spectrophotometer. The obtained value, calculated after the assessment of the absorbance at a wavelength of 750 nm, was equal to 1.12 × 10^−3^ g gallic acid derivative per gram of empty hydrogel and of 4.8 × 10^−3^ g gallic acid derivative per gram of loaded hydrogel, probably due to the presence of phloretin, a type of natural phenol.

### 2.6. DPPH and ABTS Radical Scavenging Activity Assay

The free radical scavenging capacity of hydrogels was assessed using ABTS^+^ and DPPH radical scavenging assay. The color change in the reaction solution showed that both hydrogels, empty and loaded with phloretin, can scavenge free radicals (Figure 5). As shown in Figure 5a, the ABTS^+^ free radical scavenging activity was higher for the loaded hydrogel than for the empty one and equal to 61%. As shown in Figure 5b, empty hydrogel showed a reduced DPPH clearance compared to the DPPH clearance of loaded hydrogel due to the increase in antioxidant content. Furthermore, Figure 4b showed the concentration dependence of the radical scavenging ability of both hydrogels, and the concentration of the hydrogel for optimal clearance rate was consistent with the concentration for ABTS clearance.

### 2.7. Antioxidant Activity Evaluation

The ability of the gallic acid-hydrogel loaded with phloretin and empty, and subjected to release studies in SGF and SIF, to inhibit lipid peroxidation induced by *t*-BOOH, a source of free radicals, was examined in rat liver microsomal membranes for a 240 min incubation period. The antioxidant activity of the hydrogel was time-dependent and preserved over time. In fact, as can be seen from Figure 6, control MDA levels increased significantly, indicating the existence of severe oxidative stress [22]. However, after treatment with the empty GA hydrogel, these abnormal indices tended to normalize to significantly lower in the presence of phloretin. These results suggest that the gallic acid-hydrogel containing phloretin effectively attenuated oxidative stress in relation to the synergistic antioxidant activity of gallic acid and phloretin.

## 3. Conclusions

Chronic hyperglycemic conditions can lead to a serious imbalance between ROS generation and antioxidant defense in the organs, causing oxidative damage to cellular constituents. The hydroxyl groups of naturally occurring antioxidant substances, such as gallic acid and phloretin, are able to interact with other molecules capable of generating free radicals and mediating oxidation. They can increase the translocation and expression of GLUT4 in hyperglycemic environments resulting in improved glucose consumption and tolerance. The aim of our work was to develop a gallic acid diacrylate-based hydrogel containing phloretin to facilitate its intestinal release by preventing its chemical degradation and increasing its bioavailability. The results obtained from FT-IR characterizations showed the formation of both gallic acid diacrylate and hydrogel. The gallic acid-hydrogel showed better performance at a pH of 7.4 in swelling and release studies, while the evaluation of free radical scavenging capacity using DPPH and ABTS tests showed a dependence of both hydrogels on concentration and an optimal clearance rate consistent with the concentration for ABTS clearance. The hydrogel’s antioxidant activity was preserved and resulted time dependent. Therefore, it can be speculated that this hydrogel may be useful as an innovative intestinal phloretin delivery system, also thanks to its capacity to reduce the level of ROS in the hyperglycemic environment typical of chronic diabetes.

## 4. Materials and Methods

### 4.1. Reagents

Hydrochloric acid, chloroform, diethyl ether, ethanol, isopropanol, methanol, *n*-hexane, tetrahydrofuran (THF) and sodium sulfate were purchased from Carlo Erba Reagents (Milan, Italy). Gallic acid (MW = 170.12), phloretin (MW = 274.27), acrylic acid (MW = 72.06, d = 1051 g/mL), dicyclohexylcarbodiimide (DCC), N,N-dimethylaminopyridine (DMAP), N, N-dimethylacrylamide (DMAA), *tert*-butylhydroperoxide (*t*-BOOH), trichloroacetic acid (TCA) acid, 2-thiobarbituric acid (TBA), and butylated hydroxytoluene (BHT) were purchased from Sigma-Aldrich (SGFma Chemical Co., St Louis, MO, USA). Folin–Ciocolteau and 2,2-difenil-1-picrylidrazyl (DPPH) were purchased from Sigma-Aldrich (SGFma Chemical Co., St Louis, MO, USA), while acid 2,2′-azino-bis (ABTS) was purchased from Alfa Aesar (Ward Hill, MA, USA).

### 4.2. Instruments

^1^H-NMR spectra were realized by using a Bruker VM 30 spectrophotometer (Bruker, Milano, Italy) and FT-IR spectra were obtained using a Jasco 4200 spectrophotometer Jasco, Molano, Italy). UV-Vis spectra were realized using a Jasco V-530 UV/Vis spectrophotometer (Jasco, Milano, Italy) and scanning electron microscopy (SEM) by using a JSMT 300 (Jeol, Basiglio, Milano, Italy). A Rotavapor was used to remove solvent, while a Micro Modulyo, (Edwards, Sabaudia, Latina, Italy) was used to freeze-dry samples. Thin-layer chromatography (TLC) was performed using silica gel plates 60 F254 on aluminum supplied by Merck (Milano, Italy) using UV light.

### 4.3. Animals

The animal study protocol was approved by the Italian Ministry of Health (Rome, Italy); (protocol code 700A2N.6TI, date of approval: March 2018). The animal procedures were conducted according to guidelines approved by the University Committee for Animal Welfare (OPBA) of the Department of Pharmacy, Health and Nutrition Sciences of the University of Calabria (Italy) and in compliance with the Council Directive (86/609/EEC) and Legislative Decree 26/2014 in order to apply the principles of reduction and refinement. The experiments were conducted on Wistar rats (250–300 g), obtained from Charles River Laboratories (Lecco, Italy) and they were housed in transparent polyethylene cages measuring 36 cm × 18.5 cm × 24 cm in a room with a controlled temperature (22 ± 1 °C) and with a light-dark program (lights on from 7:00 to 19:00) for at least 7 days before being used. Food and water were available. Rats were sacrificed by exposure to 4% isoflurane followed by cervical dislocation.

### 4.4. Acrylation of 3,4,5-Trihydroxybenzoic Acid with 2-Propenoic Acid

In a three-necked flask equipped with a reflux condenser and a magnetic stirrer, accurately flamed and maintained under inert atmosphere, acrylic acid (0.14 mL, 2 × 10^−3^ moles) was solubilized in 30 mL of dry tetrahydrofuran (THF). After the addition of DCC (0.42 g, 2 × 10^−3^ moles) and DMAP (0.05 g, 4 × 10^−4^ moles), the reaction mixture was kept under stirring and heat (50 °C). After 1 h, the solution was added with gallic acid (1 g, 5.8 × 10^−3^ moles). The reaction was carried on for about 72 h and under magnetic stirring at 50 °C and monitored through thin-layer chromatography (TLC/silica gel, 7:3 chloroform-methanol eluent mixture). Dicyclohexylurea (DCU) formed during the reaction; it was removed via filtration. The reaction solvent was removed via evaporation under reduced pressure. The obtained product, which was gelatinous in consistency and yellow in color, was purified via column chromatography on silica gel (eluent mixture: chloroform). The purified product was characterized through FT-IR spectrophotometry, GC-MS, and ^1^H-NMR.

### 4.5. Preparation of the Diacrylate Gallate Hydrogel

Gallic acid diacrylate (0.05 g, 1.8 × 10^−4^ mol) was solubilized in an aqueous solution of NH_3_/urea (2.5 mL) to which N, N-dimethylacrylamide comonomer (0.052 g, 5.2 × 10^−4^ mol) and ammonium persulfate radical initiator (0.12 g, 5.2 × 10^−4^ mol) were added. The resulting solution was kept under stirring and was heated (60 °C) until the hydrogel was formed. The latter, washed with distilled water, frozen and lyophilized, was carefully characterized using FT-IR, which showed the complete disappearance of the typical bands of the acrylic group at 985 cm^−1^ and 905 cm^−1^ and the appearance of a new carbonyl stretching band at 1640 cm^−1^ [23,24].

### 4.6. Swelling Studies

The swelling behavior of the hydrogels, in accordance with a procedure reported in the literature, ref. [25], was evaluated at three different pHs (1.2, 6.8 and 7.4) which simulated the conditions typical of the gastro-intestinal tract, and at predetermined time intervals (1 h, 2 h, 3 h, 4 h). In total, 0.05 g aliquots of the hydrogel were placed in glass filters, previously weighed, and immersed in beakers containing buffer solutions of different pHs (1.2 to mimic the acid environment of the stomach and phosphate buffer at 6.8 and 7.4 to mimic the intestine ones). At predetermined time intervals the excess water was removed from the filters through percolation. Subsequently, the filters were centrifuged at 8000 rpm for 5 min and weighed. The swelling degree (% α) was calculated by Equation (1):(1)$$\alpha \%=\frac{Ws-Wd}{Wd}\times 100$$where *Ws* is the weight of the swelled hydrogel and *Wd* is the weight of the initial one (dry). The weights, measured at all time intervals, have been used to calculate the hydrogel swelling degree. The swelling was evaluated for the first two hours at pH 1.2, then the pH was adjusted to 7.4 and at this pH the swelling degree was estimated from the third hour onwards [26].

### 4.7. Impregnation of the Hydrogel with Phloretin

The hydrogel was loaded with phloretin using a drug solution of water/ethanol 8/2. The whole was left under constant stirring at 37 °C in a water bath for 72 h. The solution was analyzed, after filtration, through UV-Vis (λ = 288 nm, ε = 3201.4 mol·dm^−1^·cm^−1^). This allowed us to calculate the drug loading efficiency (LE%) of the hydrogel through the following equation (Equation (2)):(2)$$LE\%=\frac{Ci-Cf}{Cf}\times 100$$

*Ci* is the initial drug concentration in solution, while *Cf* is the drug concentration in solution after the loading study. *LE*% was equal to 80%.

### 4.8. In Vitro Release Studies

The in vitro release of phloretin, simulated gastric fluid (SGF, hydrochloric acid pH 1.2), simulated intestinal fluid (SIF, phosphate buffer pH 6.8) and simulated colon fluid (SCF, phosphate buffer pH 7.4), respectively, were placed into a dialysis tubing, at different time intervals (1 h, 2 h, 3 h, 6 h, 12 h). The temperature of dissolution medium (900 mL) was maintained at 37 ± 0.5 °C throughout the test. At predetermined time intervals of 1 h, 2 h, 3 h, 6 h, 12 h, a 5 mL sample was withdrawn and was replaced with the fresh medium to maintain sink condition. The samples were analyzed using UV-visible spectrophotometer. The phloretin concentration in the different solutions was monitored by the use of UV-Vis spectrophotometry at 288 nm. The data were calculated in terms of drug release percentage [27].

### 4.9. Determination of Total Phenolic Content

The Folin–Ciocolteau method was used to evaluate the total phenolic content. A total of 150 mg of Folin–Ciocolteau reagent was added to 150 mg of the hydrogel and left in the dark for 6 min. Then, 3 mL of Na_2_CO_3_ at 20% and 5.5 mL of distilled water were added and incubated in the dark for 30 min. The sample was centrifuged for 10 min and the absorbance was measured spectrophotometrically at 750 nm. A calibration curve using GA was carried out and the total phenolic content were expressed as GA equivalents (GAE) [28].

### 4.10. DPPH Radical Scavenging Activity Assay

The ability of prepared hydrogels to act as radical scavengers was considered using DPPH reaction. Briefly, different concentrations of empty and phloretin-loaded hydrogels (60, 110, 160, 210 mg) were incubated with 3 mL of 0.25 mM DPPH ethanolic solution at room temperature and in the dark. After 30 min, absorbance measurements were made at 517 nm with a UV-vis spectrophotometer against ethanol as blank. The scavenging activity of the DPPH radical was calculated according to the following equation:$$Scavenging\;activity\;(\%)=\frac{\;(A{}^{\circ}-A1)}{A{}^{\circ}}\times 100$$where *A*° is the absorbance of control (blank) and *A*1 is the absorbance in presence of hydrogel. The results were expressed as means ± SD [28].

### 4.11. ABTS Radical Scavenging Assay

A stock solution of 7 mM ABTS was mixed with potassium persulfate 2.45 mM and stirred in the dark for 12 h in order to generate the ABTS free radical. Before use, the solution was diluted with ethanol until an absorbance of 0.7 was reached. Then, different concentrations of empty and phloretin-loaded hydrogels (60, 110, 160, 210 mg) were added to 3 mL of the ABTS solution and after 6 min the absorbance was evaluated at 734 nm using a UV spectrophotometer. Finally, ABTS scavenging activity was calculated, using the equation:$$\%\;\mathrm{inhibition}\;(\%)=\frac{\;(A{}^{\circ}-As)}{A{}^{\circ}}\times 100$$
where, *As* is the absorbance of the sample at 734 nm and *A*° is the control. All tests were realized in triplicate and the results expressed as means ± SD [28].

### 4.12. Evaluation of the Antioxidant Activity

Liver microsomes were prepared from Wistar rats via tissue homogenization with 5 volumes of ice-cold 0.25 M sucrose containing 5 mM Hepes, 0.5 mM EDTA, pH 7.5 in a Potter–Elvehjem homogenizer. Microsomal membranes were isolated by the removal of the nuclear fraction at 8000× *g* for 10 min and removal of the mitochondrial fraction at 18,000× *g* for 10 min. The microsomal fraction was sedimented at 105,000× *g* for 60 min, and the fraction was washed once in 0.15 M KCl and collected again at 105,000× *g* for 30 min. The membranes, suspended in 0.1 M potassium phosphate buffer, pH 7.5, were stored at −80 °C. Aliquots of hydrogels in the range of 5–6 mg/mL were added to the microsomes. The microsomes were gently suspended using a Dounce homogenizer, and then the suspensions were incubated at 37 °C in a shaking bath under air in the dark, in both the absence and the presence of *tert*-BOOH. Malondialdehyde (MDA) was extracted and analyzed as indicated. Briefly, aliquots of 1 mL of microsomal suspension (0.5 mg proteins) were mixed with 3 mL 0.5% TCA and 0.5 mL of TBA solution (two parts 0.4% TBA in 0.2 M HCl and one part distilled water) and 0.07 mL of 0.2% BHT in 95% ethanol. Hydrogel samples were then incubated in a 90 °C bath for 240 min. After incubation, the TBA–MDA complex was extracted with 3 mL of isobutyl alcohol. The absorbances of the extracts were measured using UV spectrophotometry at 535 nm and the results were expressed as nmol per mg of protein [29]. 

### 4.13. Statistical Analysis

All quantitative data were expressed as means ± standard deviations. Differences between means were analyzed for statistical significance using Student’s *t*-test. In this study, *p*-values less than 0.05 were considered statistically significant.

## Data Availability

Data are contained within the article.

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
