# Peer review of "Gallic Acid-Based Hydrogels for Phloretin Intestinal Release: A Promising Strategy to Reduce Oxidative Stress in Chronic Diabetes"

_molecules, 2024, doi:10.3390/molecules29050929_

Round 1
Reviewer 1 Report
Comments and Suggestions for Authors
1. Can you add more direct evidence based on in vivo animal experiments demonstrating that gallic acid-based hydrogels could reduce glucose concentration in postprandial blood sugar?
2. What is the best dose of gallic acid-based hydrogels that could have the best effect on glucose control especially in animal experiments?
3. Can you add more illumination of the mechanism how the hydrogels controls glucose in blood based on experiments level?
Comments on the Quality of English LanguageMinor modification
Author Response
Review 1
Can you add more direct evidence based on in vivo animal experiments demonstrating that gallic acid-based hydrogels could reduce glucose concentration in postprandial blood sugar?
- What is the best dose of gallic acid-based hydrogels that could have the best effect on glucose control especially in animal experiments?
- Can you add more illumination of the mechanism how the hydrogels controls glucose in blood based on experiments level?
In vivo experiments on animals were not conducted because unfortunately we do not work with animals, and we have no authorisation from the ethics committee to carry out such work. In addition, the focus of the work is purely on the antioxidant action of the hydrogel in a hyperglycaemic environment where the presence of reactive oxygen substances increases oxidative stress. We are therefore unable to answer the reviewer's question two and three.
Reviewer 2 Report
Comments and Suggestions for Authors
The article is about developing and characterizing a gallic acid-based hydrogel for oral delivery of phloretin, a natural compound with antidiabetic and antioxidant properties. The article is well-written and organized, with a clear introduction, methods, results, and discussion sections. The article provides sufficient background information on the rationale and objectives of the study, as well as the relevant literature on the properties and applications of phloretin and gallic acid. The article also describes the experimental procedures and techniques used to synthesize and characterize the hydrogel, such as FT-IR spectroscopy, swelling tests, release studies, and antioxidant assays. The article presents and analyzes the results logically and coherently, highlighting the main findings and implications of the study. The article discusses the potential advantages and limitations of hydrogel as a novel delivery system for phloretin and the future directions and challenges for further research. The article concludes with a concise and informative summary of the main points and contributions of the study.
The article is generally well-referenced, with appropriate citations to support the claims and arguments made by the authors. However, minor improvements could be made to enhance the quality and clarity of the article. Here are some suggestions for revision:
1. The abstract should be more concise and focused, avoiding unnecessary details and repetitions.
- The introduction should provide more context and motivation for the study, explaining why phloretin is a promising candidate for the treatment of diabetes, and what are the current challenges and limitations of its oral delivery. The introduction should also state the specific aims and hypotheses of the study and how they address the existing gaps in the literature.
- It is advisable to highlight the significance of synthetic endeavors in the advancement of antidiabetic medication. Within this context, it is recommended to underscore the significance of iminosugars and sugar derivatives as potent antidiabetic agents. To support this assertion, referencing the subsequent pertinent articles in the introduction section is suggested: i) https://doi.org/10.1002/anie.202217809 ii) Compain, P.; Martin, O. R. Iminosugars: From synthesis to therapeutic applications; Wiley-VCH:New York, 2007; pp 187−298 and iii) https://doi.org/10.24820/ark.5550190.p011.809.
- The methods section should provide more details on the materials and sources used for synthesizing the hydrogel and the instruments and settings used for the characterization techniques. The methods section should also specify the statistical methods and software used for the data analysis and the level of significance and confidence intervals used for the hypothesis testing.
- The results section should include more graphical representations of the data, such as tables, charts, and figures, to facilitate the visualization and comparison of the results. The results section should also avoid repeating the numerical values already shown in the graphs and instead focus on describing the trends and patterns observed in the data. The results section should also report the standard deviations or errors of the measurements and the tests' p-values or significance levels to indicate the results' reliability and validity.
- The discussion section should provide more interpretation and explanation of the results, relating them to the objectives and hypotheses of the study and comparing them with the previous studies and findings in the literature. The discussion section should also discuss the implications and applications of the results for developing a novel delivery system for phloretin, the limitations and challenges of the study, and suggestions for future research.
- The conclusion section should summarize the main points and contributions of the study, highlighting the novelty and significance of the findings and the potential impact and benefits of hydrogel for the treatment of diabetes. The conclusion section should also provide a clear and concise answer to the research question or problem posed in the introduction and refrain from introducing new information or arguments not discussed in the article.
- The references section should follow a consistent and standard format, including the authors, year, title, journal, volume, issue, pages, and DOI. The references section should also avoid using secondary sources, such as websites or books, and instead use primary sources, such as peer-reviewed articles or reports, whenever possible. The references section should also ensure that all the text’s citations match the references in the list and vice versa.
Author Response
Review 2
The article is generally well-referenced, with appropriate citations to support the claims and arguments made by the authors. However, minor improvements could be made to enhance the quality and clarity of the article. Here are some suggestions for revision:
- The abstract should be more concise and focused, avoiding unnecessary details and repetitions.
The abstract has been revised and rewritten.
- The introduction should provide more context and motivation for the study, explaining why phloretin is a promising candidate for the treatment of diabetes, and what are the current challenges and limitations of its oral delivery. The introduction should also state the specific aims and hypotheses of the study and how they address the existing gaps in the literature.
It is advisable to highlight the significance of synthetic endeavors in the advancement of antidiabetic medication. Within this context, it is recommended to underscore the significance of iminosugars and sugar derivatives as potent antidiabetic agents. To support this assertion, referencing the subsequent pertinent articles in the introduction section is suggested: i) https://doi.org/10.1002/anie.202217809 ii) Compain, P.; Martin, O. R. Iminosugars: From synthesis to therapeutic applications; Wiley-VCH:New York, 2007; pp 187−298 and iii) https://doi.org/10.24820/ark.5550190.p011.809.
The introduction has been amended to take into account requests made by the reviewers. We did not emphasize the importance of iminosugars and sugar derivatives as potent anti-diabetic agents because being a scientific article and not a review, we focused our attention on the objective of the work which was to demonstrate the ability of this hydrogel to reduce oxidative stress levels typical of a hyperglycaemic environment through the synergistic action of two antioxidant substances such as gallic acid and phloretin.
- The methods section should provide more details on the materials and sources used for synthesizing the hydrogel and the instruments and settings used for the characterization techniques. The methods section should also specify the statistical methods and software used for the data analysis and the level of significance and confidence intervals used for the hypothesis testing.
The materials and methods section has been updated.
- The results section should include more graphical representations of the data, such as tables, charts, and figures, to facilitate the visualization and comparison of the results. The results section should also avoid repeating the numerical values already shown in the graphs and instead focus on describing the trends and patterns observed in the data. The results section should also report the standard deviations or errors of the measurements and the tests' p-values or significance levels to indicate the results' reliability and validity.
In the results section, graphs derived from FT-IR characterisations of the djacrylate prodrug and hydrogel have been added. In addition, as new studies were conducted to assess the polyphenol content and scavenging capacity of hydrogels (via DPPH and ABTS), the graphs relating to these activities were also included in the text.
- The discussion section should provide more interpretation and explanation of the results, relating them to the objectives and hypotheses of the study and comparing them with the previous studies and findings in the literature. The discussion section should also discuss the implications and applications of the results for developing a novel delivery system for phloretin, the limitations and challenges of the study, and suggestions for future research.
The discussion section has been expanded to include new analyses conducted on the starting materials and hydrogels obtained.
- The conclusion section should summarize the main points and contributions of the study, highlighting the novelty and significance of the findings and the potential impact and benefits of hydrogel for the treatment of diabetes. The conclusion section should also provide a clear and concise answer to the research question or problem posed in the introduction and refrain from introducing new information or arguments not discussed in the article.
The conclusion section has been updated.
- The references section should follow a consistent and standard format, including the authors, year, title, journal, volume, issue, pages, and DOI. The references section should also avoid using secondary sources, such as websites or books, and instead use primary sources, such as peer-reviewed articles or reports, whenever possible. The references section should also ensure that all the text’s citations match the references in the list and vice versa.
The references section has been updated.
Round 2
Reviewer 1 Report
Comments and Suggestions for Authors
May I know where is the discussion part?